# Isolation and characterization of lumpy skin disease virus from cattle in India

Naveen Kumar[1]*, Yogesh Chander[1], Ram Kumar[1], Nitin Khandelwal[1], Thachamvally Riyesh[1], Khushboo Chaudhary[1], Karuppusamy Shanmugasundaram[1], Sanjit Kumar[2], Anand Kumar[3], Madhurendu K. Gupta[2], Yash Pal[1], Sanjay Barua[1]*, Bhupendra N. Tripathi[1]*

1 National Centre for Veterinary Type Cultures, ICAR-National Research Centre on Equines, Hisar, India, 2 Department of Veterinary Pathology, College of Veterinary Sciences, Birsa Agricultural University, Ranchi, India, 3 Department of Veterinary Gynaecology and Obstetrics, College of Veterinary Sciences, Birsa Agricultural University, Ranchi, India

* naveenkumar.icar@gmail.com (NK); sbarua06@gmail.com (SB); bntripath1@yahoo.co.in (BNT)

## Abstract

Lumpy skin disease (LSD) has devastating economic impact. During the last decade, LSD had spread to climatically new and previously disease-free countries, which also includes its recent emergence in the Indian subcontinent (2019). This study deals with the LSD outbreak(s) from cattle in Ranchi (India). Virus was isolated from the scabs (skin lesions) in the primary goat kidney cells. Phylogenetic analysis based on nucleotide sequencing of LSD virus (LSDV) ORF011, ORF012 and ORF036 suggested that the isolated virus (LSDV/*Bos taurus*-tc/India/2019/Ranchi) is closely related to Kenyan LSDV strains. Further, we adapted the isolated virus in Vero cells. Infection of the isolated LSDV to Vero cells did not produce cytopathic effect (CPE) until the 4th blind passage, but upon adaptation, it produced high viral titres in the cultured cells. The kinetics of viral DNA synthesis and one-step growth curve analysis suggested that Vero cell-adapted LSDV initiates synthesizing its genome at ~24 hours post-infection (hpi) with a peak level at ~96 hpi whereas evidence of progeny virus particles was observed at 36–48 hours (h) with a peak titre at ~120 h. To the best of our knowledge, this study describes the first successful isolation of LSDV in India, besides providing insights into the life cycle Vero cell-adapted LSDV.

## Introduction

Lumpy skin disease (LSD) is a trans-boundary animal viral disease which causes considerable financial losses to the livestock industry. It was observed for the first time in Zambia in 1929 [1] and spread rapidly in the cattle population across African countries [reviewed in reference [2]]. Until 1984, LSD was maintained within the countries of sub-Sahara Africa [2]. The first confirmed transcontinental spread of LSD from the African to Middle-East Asian countries occurred when the disease was reported in Israel in 1989 [3]. In 2013, it was confirmed in Turkey. By 2015–16, the disease was reported in South-East Europe, the Balkans and the Caucasus [4]. Of late the disease was reported for the first time from India in November 2019 [5].

**Data Availability Statement:** All relevant data are within the manuscript and its Supporting Information files.

**Funding:** This work was supported by the Science and Engineering Research Board, Department of

Science and Technology, Government of India, Grant number CRG/2018/004747 and CVD/2020/000103 to NKu and Indian Council of Agricultural Research, Grant Number IXX11882 to SB. The funders had no role in study design, data collection and analysis, decision to publish, or preparation of the manuscript.

**Competing interests:** The authors have declared that no competing interests exist

Clinically, LSD has been reported in cattle only. The incubation period of the disease is 4–12 days. The clinical picture starts with fever (40–41.5˚C) which persists for 1–3 days [6]. This is accompanied by increased nasal and pharyngeal secretions, lachrymation, enlargement of lymph nodes, anorexia, dysgalactia, general depression and a disinclination to move [7]. The skin nodules appear within 1–2 days, which gradually become harder and necrotic thereby inducing severe discomfort, pain and lameness. In 2–3 weeks, the nodules either regress, or necrosis of the skin results in hard, raised areas (sit-fasts) clearly separated from the surrounding skin. Some of the sit-fasts may slough away, leaving a full skin thickness hole in the skin which usually gets infected by bacteria or becomes liable to myasis [8]. Some animals become extremely emaciated, and euthanasia may be warranted. Besides, the bulls may become temporarily or permanently infertile and may secrete the virus for a prolonged period. The morbidity in LSD varies from 50–100% [9]. The mortality rate is usually low (1–5%) but occasionally reported to be much higher [10].

The occurrence of LSD causes decreased milk production, loss of hide and draft. An economic loss of 20.9 million Euros was estimated in the 2016 outbreak of LSD in Balkan countries, i. e. Albania, Bulgaria, Macedonia [10]. In several African countries, LSD causes serious problems because it occurs during season, a time when draught oxen are required for cultivation of crops and thereby resulting in failure to cultivate and plant crops [11]. This constituted a serious hazard to the food security of the people in the affected areas [11].

LSD virus (LSDV) belongs to the genus capripoxvirus within the family *Poxviridae*. LSDV genome is ~151 kbp in length [12]. Two other capripoxviruses, sheeppox virus (SPV) and goatpox virus (GPV) which cause devastating disease in sheep and goats respectively, are also antigenically similar to LSDV [13]. Capripoxviruses are cross-reactive within the genus; therefore SPV- or GPV-based vaccines have been used to provide cross protection against LSDV [13–15]. LSDV is usually isolated and quantified (tissue culture infective dose; $TCID_{50}$) in primary cells. LSDV also infects Madin-Darby bovine kidney (MDBK) cells where it forms foci (multifocal areas of hyperplastic cells). These foci can be counted under microscope in an agar-overlay medium. However, a plaque assay to precisely quantify infectious LSDV is still lacking.

The recent and unprecedented spread of LSD in India and several other countries has highlighted the need for better research efforts into this rapidly emerging pathogen. Our study describes alternative cells for isolation and *in vitro* propagation of LSDV, besides providing insights on LSDV life cycle.

## Materials and methods

### Ethics statement

The study involves collection of biological specimens from cattle (field animals). Scabs and blood samples (3 ml each) were collected from the LSD suspected cattle (n = 22) as per the standard practices without using anaesthesia. College of Veterinary Sciences, Birsa Agricultural University, Ranchi (India) granted the permission to collect the biological specimens. A due consent was also taken from the farmer (animal owner) before collection of the specimens.

### Cells

Primary goat testicle (PGT) cells [16], primary goat kidney (PGK) cells [17], primary lamb testicle (PLT) cells [16] and Madin-Darby bovine kidney (MDBK) cells [18] were available at National Centre for Veterinary Type Cultures (NCVTC), Hisar and grown in Dulbecco's Modified Eagle's Medium (DMEM) supplemented with antibiotics and 10% foetal calf serum.

## Clinical specimens

The samples were collected from in and around Ranchi, Jharkhand, India (23.3441° North, 85.3096° East) which includes an organized cattle dairy farm and small dairy units in villages namely Tussum, Nagri, Khemra and Gadgaon. The first evidence of the disease was reported about 2 months prior to the sampling. Scabs from the nodular lesions were collected in 3 ml Minimum Essential Medium (MEM, transport medium) and transported on ice to the laboratory. At least 2 scabs were taken from each individual animal. The samples (scabs) were triturated to make ~10% suspension in MEM followed by filtration through 0.45 μM syringe filter and storage at -80°C until use. Serum/blood samples were also collected from infected as well as apparently healthy animals after taking a due consent from the farmers.

## Identification of the agent(s)

Typical nodular lesions on the body surface were primarily suggestive of LSD. Therefore, initially we investigated for the presence of capripoxvirus and LSDV-specific gene segments by PCR (Table 1). Briefly, total DNA was extracted from swab samples by DNeasy Blood & Tissue Kits as per the instructions of the manufacturer (Qiagen, Valencia, CA, USA) and resuspended in nuclease free water. The DNA was subjected to amplify capripoxvirus-specific and LSDV-specific DNA segments by PCR. Primers, melting temperatures and extension times for amplification of various agents are given in Table 1. For PCR amplification, each reaction tube of 20 μl contained 10 μl of Q5 High-Fidelity 2× Master Mix (New England BioLabs Inc.), 20 pmol of forward and reverse primer, and 5 μl of DNA (template). The thermocycler conditions were as follows: a denaturation step of 5 min at 98°C followed by 35 cycles of amplification [(30 sec at 98°C, 30 s at 52–55°C (Table 1) and 30–80 s (Table 1) at 72°C], and a final extension step at 72°C for 10 min. The PCR products were separated in a 1% agarose gel.

## Nucleotide sequencing

In order to further confirm the identity of the virus (LSDV/India/2019/Ranchi), ORF011, ORF012 and ORF036 encoding G-protein-coupled receptors (GPCRs), Ankyrin repeat (ANK) and RNA polymerase subunit (RPO30) respectively, were amplified by PCR, gel purified using QIAquick Gel Extraction Kit (Qiagen, Hilden, Germany) and subjected to direct sequencing using both forward and reverse PCR primers. Duplicate samples were submitted for sequencing and only high-quality sequences were deposited in GenBank database with Accession Numbers of MN967004 (ORF011), MN967004 (ORF12) and MN967004 (ORF036).

## Phylogenetic analysis

The nucleotide sequences from LSDV ORF011, ORF012 and ORF036 were edited to 1029, 622 and 579 bp fragments respectively using BioEdit version 7.0. These sequences, together with

**Table 1. Primers used to amplify capripoxvirus- and LSDV-specific gene segments.**

| Oligos | Gene/ORF | Primers | Product size (bp) | Thermocycler |
|---|---|---|---|---|
| Capripoxvirus | *P32* | Forward: 5'-GGAATGATGCCRTCTARATTC-3' | 199 | Annealing = 55°C |
| | | Reverse: 5'-CCCTGAAACATTAGTATCTGT-3' | | Extension = 30 s |
| LSDV | ORF 011 | Forward: 5'-ATGAATTATACTCTTAGYACAGTTAG-3' | 1134 | Annealing = 52°C |
| | | Reverse: 5'-TTATCCAATGCTAATACTACCAG-3' | | Extension = 80 s |
| | ORF 012 | Forward: 5'– ATGGAAAAGGAAAAATTATGTAGCG –3' | 636 | Annealing = 52°C |
| | | Reverse: 5'– TTATTGTTTGTCAAAAAAGGTGAGATTTC –3' | | Extension = 40 s |
| | ORF 036 | Forward: 5'– ATGGATGATGATAATACTAATTCATATAG –3' | 606 | Annealing = 52°C |
| | | Reverse: 5'– TTATTTTTCTACAGCTCTAAACTTCG –3' | | Extension = 40 s |

the representative nucleotide sequences of LSDV, sheeppox virus (SPV) and goatpox virus (GPV) available in the public domain (GenBank) were subjected to multiple sequence alignments using CLUSTALW (http://www.ebi.ac.uk/clustalw/index.html). Phylogenetic analysis was carried out by constructing a concatemeric Neighbor-Joining tree. Test of phylogeny was performed using Maximum Composite Likelihood method and the confidence intervals were estimated by a bootstrap algorithm applying 1,000 iterations.

### Virus isolation

For virus isolation, an aliquot of the virus (500 μl filtrate) was used to infect confluent monolayer of PGT, PGK and PLT cells for 2 h followed by addition of fresh growth medium. The cells were observed daily for the appearance of cytopathic effect (CPE).

### Adaptation of LSDV to Vero cells

In order to adapt LSDV to Vero cells, 500 μl inoculum of the PGK amplified virus was used to infect Vero cells for 4 h followed by addition of fresh growth medium and observance of CPE during sequential passage(s). Fifteen sequential passages in Vero cells were made. The growth curve and kinetics of synthesis of LSDV genome of the P15 virus was studied in Vero and MDBK cells. Virus titres were determined by $TCID_{50}$ assay and converted to estimated plaque-forming units (PFU) by the conversion $TCID_{50} \approx 0.7$ PFU as described previously [19].

### Plaque assay

Plaque was performed as per the previously described methods along with some modifications [17, 20]. The confluent monolayers of Vero cells, grown in 6 well tissue culture plates, were infected with 10-fold serial dilutions of virus for 1 h at $37^{\circ}$C, after which the infecting medium was replaced with an agar-overlay containing equal volume of 2X L-15 medium and 1% agar. Upon development of plaques, the agar-overlay was removed, and the plaques were stained by 1% crystal violet.

### Virus neutralization assay

Serum samples were initially heated at $56^{\circ}$C for 30 min to inactivate the complements. PGK cells were grown in 96 well tissue culture plates at ~90 confluency. Two-fold serum dilutions (in 50 μl volume) were made in phosphate buffered saline (PBS) and incubated with equal volume of LSDV ($10^4$ PFU/ml) for 1 h. Thereafter, virus-antibody mixture was used to infect PGK cells. The cells were observed daily for the appearance of CPE. Final reading was taken at 96 hours post-infection (hpi) for the determination of antibody titres.

### Kinetics of LSDV genome synthesis (qRT-PCR)

Vero cells, in triplicates, were infected with LSDV (Vero cell-adapted) at MOI of 5 for 2 h followed by washing with PBS and addition of fresh DMEM. Cells were scraped at various times post-infection. The levels of viral DNA in the infected cells were quantified by qRT-PCR. Viral RNA/DNA Purification Kit (Thermo Scientific, Vilnius, Lithuania) was used for extraction of viral DNA from the infected cell lysate. qRT-PCR was carried out with a 20 μl reaction mixture containing LSDV ORF036 or specific primers (Table 1), template and Sybr green DNA dye (Promega, Madison, USA). Thermal cycler conditions were as follows: a denaturation step of 5 min at $94^{\circ}$C followed by 40 cycles of amplification (30 s at $94^{\circ}$C, 30 s at $52^{\circ}$C, and 40 s at $72^{\circ}$C). The levels of LSDV DNA, expressed as threshold cycle (*Ct*) values, were normalized with β-actin gene to determine relative fold-change in copy number of viral DNA [21].

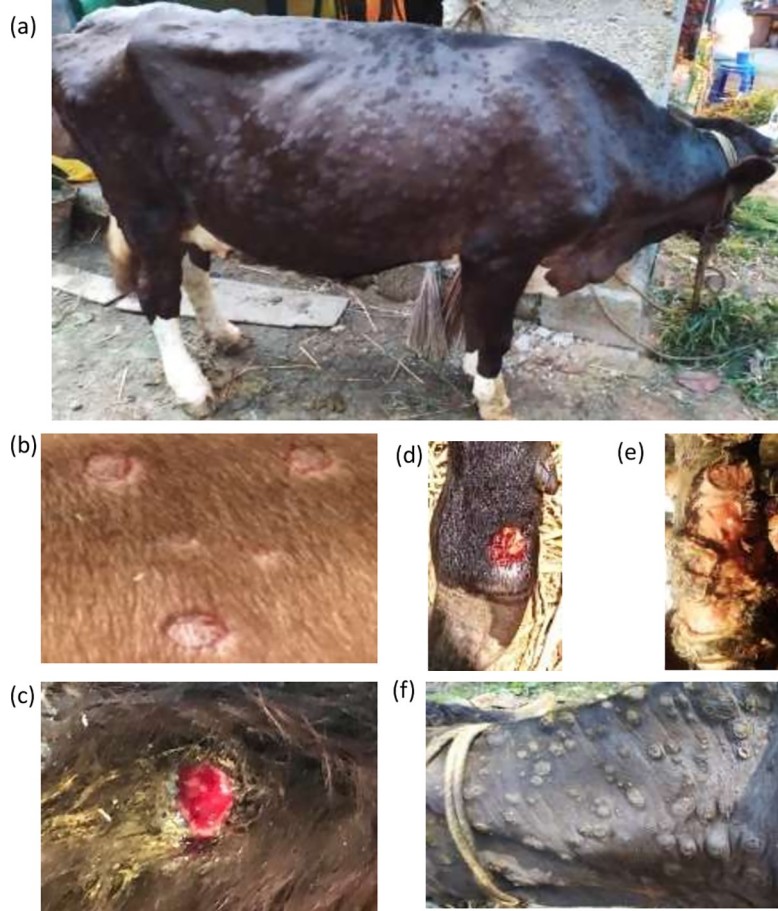

**Fig 1. Clinical findings.** (a) Nodules all over the body surface in cattle (b) Nodules are circumscribed, round, slightly raised, firm, painful and are 1–3 cm in size. (c) Ruptured nodules that created a deep-seated wound. (d). Wounds invaded by secondary bacterial infection leading to suppuration and sloughing. (e) Extensive lesions in the fetlock region extending up to the underlying subcuitis and muscle. (f) Necrosis of the skin nodule resulting in hard, raised areas "sit-fasts".

## Results

### Clinical findings

In the initial stages, the affected animals showed fever (up to 41.5°C) which persisted for 3–4 days which was followed by oedema (swelling) of legs, enlarged lymph node, lameness and anorexia. The most prominent clinical finding was the appearance of skin nodules all over the body surface (Fig 1A) immediately after the febrile stage. The nodules were well circumscribed, round, slightly raised, firm, painful and were 1–3 cm in size (Fig 1B). Some of the nodules ruptured to create a deep-seated wound (Fig 1C and 1D). Wounds were frequently invaded by secondary bacterial infections which led to extensive suppuration and sloughing. The lesions were particularly extensive in the fetlock region, extended up to the underlying subcutis and muscle (Fig 1E). Some of the nodules regressed and in some the necrosis of the skin resulted in hard, raised areas (sit-fasts) clearly separated from the surrounding skin (Fig 1F). Nasal discharge was apparent only in few animals. Few pregnant animals aborted. Morbidity was ~8% in the organized dairy farm (college dairy farm) and up to ~25% in small dairy units in the villages. Most of the animals recovered except few which died due to extensive lesions, anorexia

and emaciation. During the course of writing this manuscript, the disease has spread into several states in India including Kerala, Tamil Nadu, Andhra Pradesh, Telangana, Odisha, Jharkhand, West Bengal, Assam, Chhattisgarh, Maharashtra and Madhya Pradesh.

## Identification of the agent

In order to demonstrate the etiological agent, DNA was extracted from the scabs and subjected to amplification of capripoxvirus-specific and LSDV-specific DNA segments by PCR. An amplification of 199 bp DNA fragment with capripoxvirus-specific primers indicated the presence of capripoxvirus in the scabs (S1 Fig). All except two scab specimens were positive for capripoxvirus genome (Table 2). However, viremia could not be detected; neither in the clinically affected animals with skin nodules nor in the healthy in-contact susceptible animals (Table 2). Collectively, based on characteristic clinical signs, and amplification of capripoxvirus- and LSDV-specific genome, the identity of the virus was confirmed as LSDV.

## Reactivity of LSDV to the sera from LSDV-infected animals

Serum samples from all the clinically affected animals that showed presence of skin nodules and some in-contact susceptible animals were also subjected for determination of anti-LSDV antibody titre in a virus neutralization assay. Clinically affected animals showed an antibody titre of 1:64 to 1:1024 (Table 2). However, few healthy in-contact susceptible animals were also found positive for anti-LSDV antibodies suggesting subclinical infection (Table 2).

**Table 2. Detection of virus and virus-specific antibodies in clinically affected and in-contact healthy animals.**

| S. No. | Place | Skin nodules | LSDV genome (scab) | LSDV genome (blood) | Antibody titre |
|--------|-------|--------------|--------------------|--------------------|----------------|
| 1 | Tussum | (+) | (+) | (-) | 1:64 |
| 2 | Tussum | Healthy* | NA | (-) | 1:16 |
| 3 | Tussum | (+) | (-) | (-) | 1:128 |
| 4 | Tussum | (+) | (+) | (-) | 1:32 |
| 5 | Tussum | (+) | (+) | (-) | >1:1024 |
| 6 | Tussum | Healthy* | NA | (-) | 1:128 |
| 7 | Nagri | (+) | (+) | (-) | >1:1024 |
| 8 | Nagri | (+)# | (-) | (-) | 1:128 |
| 9 | Khemra | (+)# | (+) | (-) | 1:512 |
| 10 | Khemra | (+)# | (+) | (-) | 1:512 |
| 11 | Gadgaon | (+) | (+) | (-) | 1:512 |
| 12 | Gadgaon | (+)# | (-) | (-) | >1:1024 |
| 13 | Gadgaon | Healthy* | NA | (-) | <1:4 |
| 14 | Gadgaon | (+)# | (+) | (-) | 1:128 |
| 15 | Ranchi | (+) | (+) | (-) | 1:128 (>1:1024$) |
| 16 | Ranchi | (+) | (+) | (-) | >1024 |
| 17 | Ranchi | (+) | (+) | (-) | 1:32 |
| 18 | Ranchi | Healthy* | NA | (-) | <1:4 |
| 19 | Ranchi | Healthy* | NA | (-) | <1:4 |
| 20 | Ranchi | (+) | (+) | (-) | 1:64 |
| 21 | Ranchi | Healthy* | NA | (-) | 1:128 |

*In contact susceptible animals

#Lesions towards recovery

$ One month after initial sampling; NA: Scabs not collected

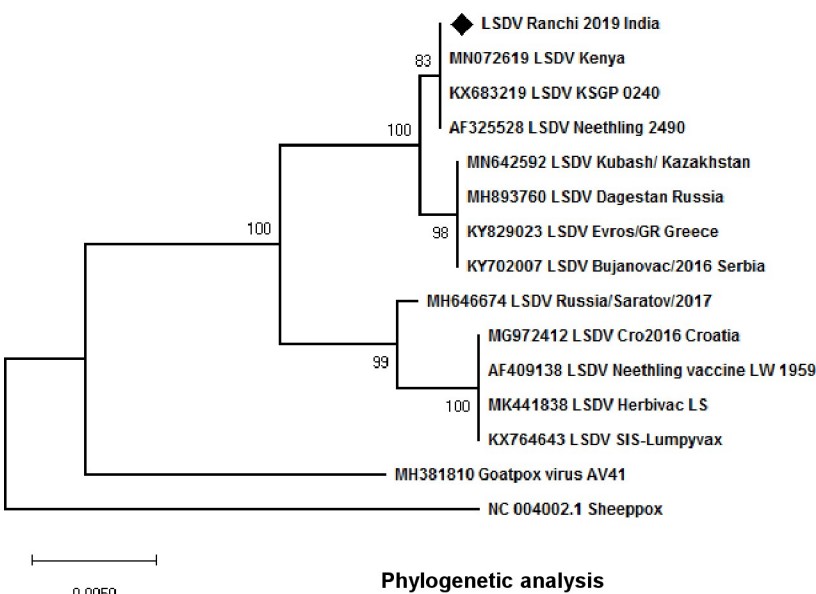

**Fig 2. Phylogenetic analysis: ORF011, ORF012 and ORF036 of LSDV/India/2019/Ranchi encoding GPCRs, Ank and RPO030 proteins respectively, were amplified by PCR and subjected to direct nucleotide sequencing.** The sequences were edited to 1029, 622 and 579 bp fragments respectively using BioEdit version 7.0. For evolutionary analysis, corresponding sequences of other LSDV strains, SPV stains and GPV strains were retrieved from GenBank. Phylogenetic analyses were carried out by constructing a concatemeric Neighbor-Joining tree. Test of phylogeny was performed using Maximum Composite Likelihood method and the confidence intervals were estimated by a bootstrap algorithm applying 1,000 iterations. The tree is drawn to scale, with branch lengths in the same units as those of the evolutionary distances used to infer the phylogenetic tree.

## Phylogenetic analysis

In order to compare and determine the phylogenetic relationship of LSDV/India/2019/Ranchi with other LSDV/SPV/GPV strains, ORF011, ORF012 and ORF036 were amplified by PCR and subjected to direct sequencing. The sequences were edited to 1029, 622 and 579 bp and deposited to GenBank with Accession Number of MN967004 (ORF011), MN967004 (ORF12) and MN967004 (ORF036). These sequences together with the corresponding nucleotide sequences from other LSDV/SPV/GPV strains retrieved from the GenBank were used to prepare a consensus linear phylogenetic tree (Fig 2). Nucleotide sequences of LSDV/India/2019/Ranchi showed highest similarity to the Kenyan LSDV strains.

## Virus isolation

The virus recovered from the scabs was used to infect PGT, PGK and PLT cells. One scab sample positive for LSDV-specific gene segment in PCR was considered for virus isolation. Infection of the clinical specimen did not reveal any CPE up to 7 days post-infection in any of the cell lines tested except in PGK cells which showed cell shrinking but no clear CPE. Thereafter, the infected cells were freeze-thawed twice and the resulting supernatant (called first blind passage) was used to infect fresh cells. Upon second blind passage, PGK cells started showing shrinking at 48–72 hpi and a clear CPE, characterized by cell rounding, ballooning and degeneration was evident at 96–120 hpi (S2A Fig). The other cell types did not produce any CPE in second blind passage, even up to 5–7 days post infection (dpi). PLT cells produced CPE in third blind (S2A Fig) passage whereas PGT cells did not show CPE even up to the fourth blind passage, suggesting PGK cells are the most sensitive cells for LSDV isolation, among the three

primary cells employed in this study. The virus was amplified in PGK cells at a titre of ~$10^7$ PFU/ml. The virus was deposited in the repository at NCVTC with an Accession Number VTCCAVA288 and named as LSDV/*Bos taurus*-tc/India/2019/Ranchi.

## Adaptation to Vero cells

Primary cells usually do not survive for long time. Therefore, it is desirable to adapt the virus in established cell lines. The clinical specimens did not produce any CPE in Vero cells at least up to 3rd blind passages. For adaptation to Vero cells, 500 μl inoculum of the PGK amplified virus was used to infect Vero cells but no CPE was observed up to 8 days post-infection after which the cells were freeze-thawed twice and used to re infect fresh Vero cells. CPE characterized by cell clustering (foci) (S2B Fig) was observed at 6 days post-infection on 4th passage (P4) in Vero cells. However, P4 virus did not form plaques. At P15, rather than producing only foci, the virus began producing a clear CPE, characterized by clustering, cell rounding and degeneration (S2B Fig). Unlike P4, P15 virus also formed plaques but were of small size (S2B Fig).

## LSDV life cycle

Little is known about the life cycle of LSDV. In order to determine the length of LSDV life cycle, Vero/MDBK cells were infected with LSDV (Vero cell-adapted) and the progeny virus particles released in the infected cell culture supernatant at various time points post-infection were quantified. As shown in Fig 3A no significant progeny virus particles were observed at 6 hpi and 24 hpi (Vero cells). The small amount of virus particles at these time points represents the input used to infect cells which remained attached with the cells/flask even upon washing. However, a sudden increase in the viral titres (progeny virus particles) was observed at 36 hpi which further increased from 48 hpi to 120 hpi finally becoming stable at 144 hpi onward and then declining at 192 hpi. The kinetics of virus replication was somewhat similar in MDBK cells, however the first evidence of progeny virus particles in infected cell culture supernatant was observed at 48 hpi. In addition, Vero cell-adapted LSDV in MDBK cells did not attain a stationary phase even up to 192 hpi and the overall viral titres were ~10-fold lower compared to the Vero cells.

In order to provide insights on the kinetics of LSDV (Vero cell-adapted) genome synthesis, Vero/MDBK cells were infected with LSDV (Vero cell-adapted) and the viral DNA was quantified in the infected cells (pellet) at various time points post-infection. As shown in Fig 3, the levels of LSDV DNA were comparable at various time points in the lag phase (at 6 hpi, 12 hpi, 18 hpi and 24 hpi), in both Vero (Fig 3B) and MDBK cells (Fig 3C). However, the levels of viral DNA progressively increased from 24 hpi to 96 hpi and finally stabilizing at 120 hpi (Fig 3B and 3C). Interestingly, higher LSDV particles adsorbed in Vero cells as compared to MDBK cells (Ct values in were in the range of 19 and 22 in Vero and MDBK cells respectively, Fig 3D). Besides, the stationary phase in Vero cells was achieved at ~72 hpi as compared to >120 hpi in MDBK cells. The experiment was not followed after 120 hpi because most of the MDBK cells died at this stage.

## Discussion

For several decades, lumpy skin disease was restricted to Africa wherein it led to several devastating pandemics in several countries, thereby threatening food security and consequently increasing poverty [22, 23]. Since the year 2000, it spread to several countries of the Middle East and was confirmed in Turkey in 2013. In 2015–16, the disease was reported in several European countries, viz; Bulgaria, Macedonia, Serbia, Kosovo, Montenegro and Albania [4,

**Kinetics of virus production (supernatant)**

**Kinetics of LSDV DNA synthesis (cell pellet)**

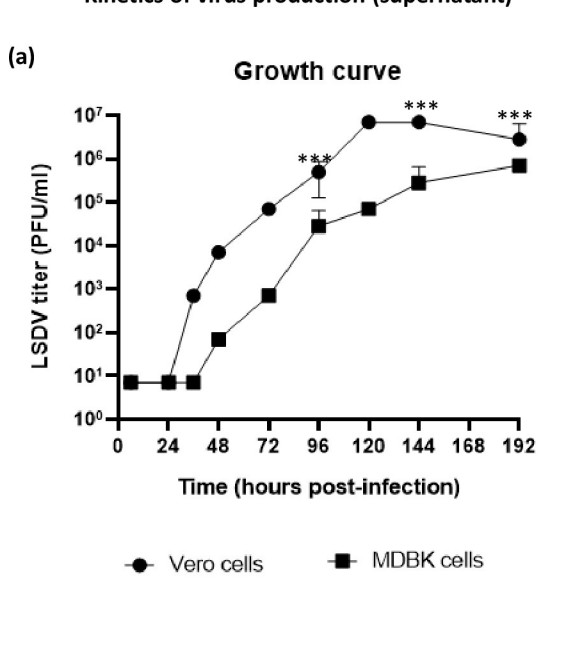

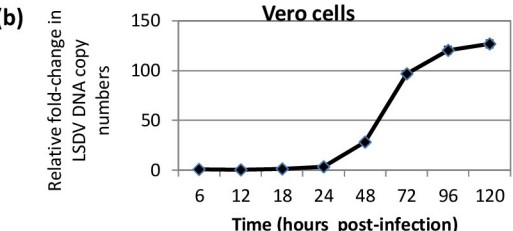

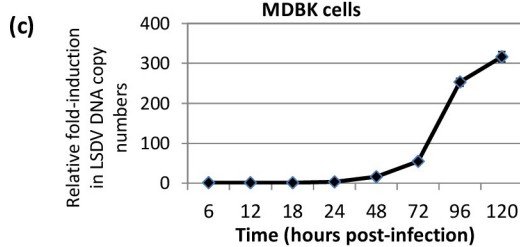

**(d)**

| | Ct values (±SD) | |
|---|---|---|
| | **Vero cells** | **MDBK cells** |
| **6 h** | 19.60±0.22*** | 22.17±0.27 |
| **12 h** | 20.38±0.15** | 22.59±0.04 |
| **18 h** | 19.25±0.06** | 21.83±0.04 |
| **24 h** | 17.74±0.13*** | 20.46±0.08 |
| **48 h** | 14.78±0.20*** | 18.16±0.07 |
| **72 h** | 13.00±0.09*** | 16.39±0.09 |
| **96 h** | 12.68±0.02** | 14.18±0.47 |
| **120 h** | 12.61±0.17* | 13.86±0.06 |

*=P<0.05, **=P<0.01, ***= P<0.001,

**Fig 3. LSDV life cycle.** Vero/MDBK cells, in triplicates, were infected with Vero cell-adapted LSDV at MOI of 5 for 2 h followed by washing with PBS and addition of fresh DMEM. *Growth curve.* Supernatant was collected from the infected cells at indicated time points and quantitated by determination of TCID$_{50}$ (expressed as PFU/ml) in Vero cells. Values are means ± SD and representative of the result of at least 3 independent experiments. Growth curve in Vero and MDBK cells **(a)** is shown. *Kinetics of viral genome synthesis.* Cells were scraped at indicated times points and the levels of viral DNA/β-actin (house-keeping control) gene in the infected cells (pellet) were measured by qRT-PCR. The levels of LSDV DNA, expressed as threshold cycle (*Ct*) values, were normalized with β-actin to determine relative fold-change in copy number of viral DNA. Kinetics of DNA synthesis in Vero **(b)** and MDBK cells **(c)** is shown. To directly compare the levels of viral DNA in Vero and MDBK cells at different times post infection, Ct values are also shown **(d)**. Values are means ± SD and representative of the result of at least 3 independent experiments. Pair-wise statistical comparisons in the viral titres/Ct values between Vero and MDBK cells (at each time point), were performed using Student's t test. * = P<0.05, ** = P<0.01, *** = P<0.001. The differences in the viral titres between Vero and MDBK cells are apparently significant at each time point except the lag phase (up to24 hpi). However, the statistical calculation is only possible at certain time points viz; 96 hpi, 144 hpi and 192 hpi where there are variations in the biological replicates.

12]. LSD was reported for the first time from India in 2019 [5]. The disease has already spread to several states *viz*; Kerala, Tamil Nadu, Andhra Pradesh, Telangana, Odisha, Jharkhand, West Bengal, Assam, Chhattisgarh, Maharashtra and Madhya Pradesh of the country and has caused considerable economic losses to the livestock industry. This recent and unprecedented spread of LSDV in India and several other countries has highlighted the need for better research efforts into this rapidly emerging pathogen. India is currently lacking in the reagents required to develop diagnostic/therapeutic/prophylactic reagents, besides lacking established laboratories for LSDV research.

Clinically, all classical symptoms of LSD *viz*; fever, generalized skin nodules, enlargement of lymph nodes, anorexia, oedema of legs and lameness [6] were observed in most of the cases we observed in the outbreak in Ranch (India). Disease was not observed in buffaloes; however, a deer exhibited skin nodules. The morbidity was ~25% without any significant mortality which

is in agreement with the previous findings [9, 10]. Presence of anti-LSDV neutralizing antibodies in some of the healthy in-contact susceptible animals (Table 2) suggests that certain animals may undergo subclinical infection in a typical LSD outbreak. Most of the collected scabs revealed the presence of LSDV genome (Table 2), however viremia was not apparent in any animal. The duration of viremia in LSDV varies between 1 to 10 days, though the virus may survive in the skin lesions for 4–6 months or longer [4]. It is quite possible that the viremic phase would have been over by the time the samples were collected.

Upon phylogenetic analyses, LSDV/India/2019/Ranchi showed the highest similarity to the Kenyan LSDV strains. The only previous study on LSDV from India also reported that Indian LSDV strains (from Odisha state of India) were closely related with Kenyan LSDV strains [5], suggesting a single LSDV strain is circulating in the country. The disease is primarily transmitted mechanically by arthropod vectors [24] and is capable of spreading across countries or even across continents by the movement of live animals [22]. It is likely that the disease could have been introduced in India by way of import of animals or animal products from Africa. There is potential for further geographic spread of LSD which necessitates increased surveillance. Although the virus isolated in this study was ~100% identical with the Kenyan LSDV strains, its complete genetic characterization (whole genome sequencing) as well as its ability to produce clinical disease needs to be further elucidated which is beyond the scope of this manuscript.

Primary cells of ovine, caprine and bovine origin are usually employed for LSDV isolation [4, 25–28]. However, primary bovine dermis and PLT cells are considered to be the most susceptible [4]. In this study, PGK cells exhibited CPE in the very first passage. PLT cells showed CPE in third blind passage whereas PGT cells did not exhibit any CPE even up to the fourth blind passage. Taken together, it can be concluded that the PGK cells are the most sensitive cells for LSDV isolation among the three primary cell lines employed. However, primary cells are prone to contamination, time consuming and expensive to produce and are not considered appropriate with current efforts to reduce the use of animals in science. MDBK cells have been commonly used for the *in vitro* propagation of LSDV at high titres [28, 29]. Some studies also suggest the use of MDBK cells for LSDV isolation [28] but it seems uncertain whether these are more sensitive than the primary cells for virus isolation.

LSDV quantitation is currently based on determination of $TCID_{50}$ in primary cells [26]. However, $TCID_{50}$ may not represent the accurate virus titre as manual observation of CPE under microscope could lead to deviations in the results partly due to the possible subjective (eye) effect. Emerging evidences also suggest use of MDBK cell line for propagation of LSDV [29–32]. A focus forming assay (in MDBK cells) has been described for LSDV [25, 29] but precise counting of foci under microscope is a tedious task. Although MDBK cells are considered sensitive for *in vitro* propagation of LSDV [28, 29], like in other cell types, virus does not produce plaques in MDBK cells. We for the first time adapted the LSDV in Vero cells by presuming that the LSDV may form plaques in an alternative cell line. No CPE could be observed upon immediate infection of LSDV (isolated in PGK cells) to Vero cells. Later on, CPE characterized by cell clustering (foci), like those observed previously in MDBK cells [29] was observed at P4. However, virus did not form any plaques at this stage (P4). On subsequent passage (P15), virus started producing a clear CPE, characterized by cell clustering, rounding and degeneration. P15 virus also produced plaques but were of small size and could not be precisely counted after crystal violet staining. Vero cell-adapted LSDV grown at a titre of ~$10^7$ PFU/ml which is similar to the virus titre obtained in primary cells [25] and MDBK cells [29], suggesting their suitability for the propagation of LSDV. Further passage(s) of LSDV (P>15) and some modifications in the plaque assay protocol are underway to enhance the plaque size. Although long-term adaptation of the LSDV in Vero cells enabled it to grow at high titres in

the cultured cells, unlike primary cells, clinical specimens did not produce CPE in Vero cells (at least up to 3 passages we followed), which suggests their unsuitability for virus isolation.

LSDV has been relatively poorly studied and very little is known about its life cycle. In this study, we also provided some insights on the life cycle of cell culture adapted LSDV. In one-step growth curve analysis, LSDV (Vero cell-adapted) was shown to start synthesizing its DNA at ~24 hpi with peak level of viral DNA at ~96 hpi. Concomitantly the progeny virus particles started appearing in the infected cell culture supernatant at ~36 hpi with a peak viral titre at ~120 hpi. The kinetics of virus replication was somewhat similar in MDBK cells, however first evidence of progeny virus particles in infected cell culture supernatant was observed at 48 hpi. In addition, LSDV titres in MDBK infected cell culture supernatant did not attain a stationary phage even up to 192 hpi and overall viral titres were ~10-fold lower than Vero cells suggesting suitability of Vero cells for the propagation of Vero cell-adapted LSDV strain. Taken together, these lines of evidence in both Vero and MDBK cells suggest that the life cycle of LSDV (Vero cell-adapted) is 36–48 h in cultured cells. These finding on LSDV which were lacking in the literature are likely to facilitate our understanding about various aspects of virus replication and pathogenesis.

This study describes the first successful isolation of LSDV in India, besides providing insights into the life cycle and *in vitro* propagation of LSDV in some primary and established cell lines.

## Supporting information

**S1 Fig. Identification of LSDV.** Virus was recovered from the scabs in DMEM followed by DNA extraction and PCR to amplify capripoxvirus-specific *P32* gene.
(TIFF)

**S2 Fig.** Virus isolation *(a) Virus isolation*. An aliquot of the virus (500 μl filtrate) was used to infect confluent monolayer of PGT, PGK and PLT cells for 2 h followed by addition of fresh growth medium. The cells were observed daily for appearance of the CPE. The CPE observed in PGK, PLT and MDBK cells is shown. *(b) Adaptation to Vero cells*. An aliquot (500 μl) of the LSDV isolated in PGK cells was used to infect confluent monolayers of Vero cells for 2 h followed by followed by addition of fresh growth medium without serum. The cells were observed daily for appearance of CPE. The CPE observed at P4 and P15 is shown.
(TIFF)

**S1 Raw image.**
(PDF)

## Author Contributions

**Conceptualization:** Naveen Kumar.

**Data curation:** Naveen Kumar, Yogesh Chander, Ram Kumar, Nitin Khandelwal, Thachamvally Riyesh, Khushboo Chaudhary, Karuppusamy Shanmugasundaram.

**Formal analysis:** Naveen Kumar.

**Funding acquisition:** Naveen Kumar.

**Investigation:** Naveen Kumar, Sanjit Kumar, Anand Kumar, Madhurendu K. Gupta.

**Methodology:** Naveen Kumar, Thachamvally Riyesh.

**Project administration:** Naveen Kumar.

**Resources:** Naveen Kumar.

**Software:** Naveen Kumar, Thachamvally Riyesh.

**Supervision:** Naveen Kumar.

**Validation:** Naveen Kumar.

**Visualization:** Naveen Kumar.

**Writing – original draft:** Naveen Kumar, Thachamvally Riyesh, Yash Pal, Sanjay Barua, Bhupendra N. Tripathi.

**Writing – review & editing:** Naveen Kumar, Yash Pal, Sanjay Barua, Bhupendra N. Tripathi.

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
