## [Decision Letter · Decision Letter 0]

9 Nov 2020

PONE-D-20-31124

Isolation and characterization of lumpy skin disease virus from cattle in India

PLOS ONE

Dear Dr. Kumar,

Thank you for submitting your manuscript to PLOS ONE. After careful consideration, we feel that it has merit but does not fully meet PLOS ONE’s publication criteria as it currently stands. Therefore, we invite you to submit a revised version of the manuscript that addresses the points raised during the review process.

Please take in account the comments of the two reviewers and take care to the spelling errors in some place. As noted by the two reviewers, please provide a few more information/ explanation in the comparison of VERO and MDBK cells cultures used in this study. In addition, please check the repository status. please provide a full explanation refering to page and line for modification in the manuscript corresponding to the answers to the reviewers in your rebutal letter.

We look forward to receiving your revised manuscript.

Kind regards,

Pierre Roques, Ph.D.

Academic Editor

PLOS ONE

Journal Requirements:

2. In your Methods section, please provide additional location information of the study area, including geographic coordinates for the data set if available.

5.  We note that Figure 2 in your submission contain map images which may be copyrighted. All PLOS content is published under the Creative Commons Attribution License (CC BY 4.0), which means that the manuscript, images, and Supporting Information files will be freely available online, and any third party is permitted to access, download, copy, distribute, and use these materials in any way, even commercially, with proper attribution. For these reasons, we cannot publish previously copyrighted maps or satellite images created using proprietary data, such as Google software (Google Maps, Street View, and Earth). For more information, see our copyright guidelines: http://journals.plos.org/plosone/s/licenses-and-copyright.

5.1.    You may seek permission from the original copyright holder of Figure 2 to publish the content specifically under the CC BY 4.0 license. 

5.2.    If you are unable to obtain permission from the original copyright holder to publish these figures under the CC BY 4.0 license or if the copyright holder’s requirements are incompatible with the CC BY 4.0 license, please either i) remove the figure or ii) supply a replacement figure that complies with the CC BY 4.0 license. Please check copyright information on all replacement figures and update the figure caption with source information. If applicable, please specify in the figure caption text when a figure is similar but not identical to the original image and is therefore for illustrative purposes only.

Reviewers' comments:

Reviewer's Responses to Questions

**Comments to the Author**

1. Is the manuscript technically sound, and do the data support the conclusions?

Reviewer #1: Yes

Reviewer #2: Partly

2. Has the statistical analysis been performed appropriately and rigorously? 

Reviewer #1: N/A

Reviewer #2: No

3. Have the authors made all data underlying the findings in their manuscript fully available?

Reviewer #1: Yes

Reviewer #2: Yes

4. Is the manuscript presented in an intelligible fashion and written in standard English?

Reviewer #1: Yes

Reviewer #2: Yes

5. Review Comments to the Author

Reviewer #1: The authors describe the first successful isolation of LSDV from a natural outbreak in India. LSDV as causative agent was confirmed by clinical observation, PCR, partial sequencing and serum neutralization assay. In addition, three different primary cell lines were examined regarding their susceptibility towards LSDV, and utilization of an alternative permanent cell line (Vero cells) for propagation of LSDV was analyzed. Moreover, authors compared viral genome load and virus titer in cell culture systems at different time points post infection to gain insight in the life cycle of LSDV.

General

• l. 184: phosphate buffered saline

• l. 211: subcutis

• l. 236: LSDV/SPV/GPV

• l. 255: PGT

• l. 256: Why the MDBK cells were not used for virus isolation in comparison?

• l. 257: brackets should be deleted

• l. 295: maybe better “Serum samples from all the clinically affected animals that showed presence of…” instead of brackets

• l. 298: titer of 1:64 to 1:1024

• l. 308: “from India in 2019”

• l. 311: “(until this paper is being written)” is not necessary

• l. 352: TCID50

• l. 379: phase

• be consistent with spelling “titer” (AE) or titre (BE) during the manuscript

Figures and Tables

• Figure 2: brackets not necessary

• Figure 3: could be given as supplemental information

• Figure 5: cold be given as supplemental information

• Table 2

o footnotes (*, #) should be described

o information about how many scabs were taken from an individual and how many scabs were taken at the same farm would be helpful

o please specify scab 22 “1 month later”, is this related to an individual animal/farm (which one?) or in general?

o In place of Viremia, better: LSDV genome (blood)

o Antibody titer

o Antibody titer should be indicated as 1:xx or transferred into ND50/ml

• Figure legend Figure 2: explain red triangles

Results

• Structure outbreak and later cell culture work together – in detail:

o “reactivity of LSDV to the sera from LSDV-infected animals” should be between “Identification of the agent” and “phylogenetic analysis” as it belongs to the examination of the outbreak and results are additionally presented in Table 2

Reviewer #2: What are the main claims of the paper and how significant are they for the discipline?

• The paper describes outbreaks of LSDV in India, and contains interesting details such as the existence of serologically positive animals which display no clinical signs, morbidity data, and phylogenetic analyses of the virus isolated. This information will be important to researchers in the field and policy makers who are trying to control the current LSD epidemic in Asia.

Are the claims properly placed in the context of the previous literature? Have the authors treated the literature fairly?

• The literature is cited well, and most of the claims are properly placed in the context. However the utility and widespread use of MDBK cells to isolate and analyse capripoxviruses is not highlighted sufficiently.

Do the data and analyses fully support the claims? If not, what other evidence is required?

• The majority of the claims are well supported. The only unsupported claim is the use of Vero cells for the propagation of LSDV (line 367-370 and elsewhere). The authors clearly show that Vero cells cannot support propagation of field strains of LSDV (line 264-269). The virus had to be passaged 15 times in order for CPE and viral plaques to be detected on Vero cells. The literature, however, shows that MDBK cells do not require such virus adaptation and CPE can be detected on first passage. The authors should therefore have concluded that Vero cells are not permissive for LSDV isolation and are inferior to MDBK cells in this respect. Their conclusion that Vero cells are suitable for the propagation of LSDV is incorrect and not supported by their results.

• Figure 5C is very indistinct, can the authors provide a better quality one? The plaques are not distinguishable on the current figure

• Please provide statistical analyses for the comparisons between Vero cells and MDBK cells in Figure 6. Are the differences reported statistically significant?

• Please re-examine the legend for Figure 6. It is not correct. A and B are growth curves. C and D (not currently mentioned in the legend) are PCRs.

• Make it clear in the legend and text for Figure 6 that the data presented is for the cell culture adapted strain of LSDV, not a field strain. This is very important for diagnosticians who may read the manuscript and not realise straight away that Vero cells do not support LSDV isolation.

• Do not use PFU/ml as a synonym for TCID50, as described in the legend of Figure 6. They are very different things. If you counted plaques (or foci) then use PFU. If you examined cells for CPE, use TCID50.

PLOS ONE encourages authors to publish detailed protocols and algorithms as supporting information online. Do any particular methods used in the manuscript warrant such treatment? If a protocol is already provided, for example for a randomized controlled trial, are there any important deviations from it? If so, have the authors explained adequately why the deviations occurred?

• Not applicable

If the paper is considered unsuitable for publication in its present form, does the study itself show sufficient potential that the authors should be encouraged to resubmit a revised version?

• Not application

Are original data deposited in appropriate repositories and accession/version numbers provided for genes, proteins, mutants, diseases, etc.?

• The LSDV strain has been deposited in a repository (lines 261-262) although this could not be confirmed on the website of the depository

Are details of the methodology sufficient to allow the experiments to be reproduced?

• Yes

Is the manuscript well organized and written clearly enough to be accessible to non-specialists?

• Yes, it is very nicely written.

6. PLOS authors have the option to publish the peer review history of their article (what does this mean?). If published, this will include your full peer review and any attached files.

Reviewer #1: No

Reviewer #2: No

---

## [Author Response · Author response to Decision Letter 0]

26 Nov 2020

Journal Requirements:

Revised manuscript meets PLOS ONE's style requirements

2. In your Methods section, please provide additional location information of the study area, including geographic coordinates for the data set if available.

Additional geographic coordinates for the data set have been provided in the text (Line 117 in the revised manuscript).

Blot/gel image data are in supporting information [S1_Fig-Raw Image (pdf file) provided in the revised manuscript]

 Phrases containing “data not shown” have been removed

5. We note that Figure 2 in your submission contain map images which may be copyrighted. All PLOS content is published under the Creative Commons Attribution License (CC BY 4.0), which means that the manuscript, images, and Supporting Information files will be freely available online, and any third party is permitted to access, download, copy, distribute, and use these materials in any way, even commercially, with proper attribution. For these reasons, we cannot publish previously copyrighted maps or satellite images created using proprietary data, such as Google software (Google Maps, Street View, and Earth). 

Figure 2 has been deleted

Responses to the Reviewer’s comments

Reviewer #1: 

The authors describe the first successful isolation of LSDV from a natural outbreak in India. LSDV as causative agent was confirmed by clinical observation, PCR, partial sequencing and serum neutralization assay. In addition, three different primary cell lines were examined regarding their susceptibility towards LSDV, and utilization of an alternative permanent cell line (Vero cells) for propagation of LSDV was analyzed. Moreover, authors compared viral genome load and virus titer in cell culture systems at different time points post infection to gain insight in the life cycle of LSDV.

General

• l. 184: phosphate buffered saline

Corrected as suggested (Line 85 in the revised manuscript)

• l. 211: subcutis

Corrected as suggested (Line 213 in the revised manuscript).

• l. 236: LSDV/SPV/GPV

Corrected as suggested (Line 243 in the revised manuscript).

• l. 255: PGT

Corrected as suggested (Line 262 in the revised manuscript).

• l. 256: Why the MDBK cells were not used for virus isolation in comparison?

Primary cells of ovine, caprine and bovine origin are considered to be the most sensitive cells for LSDV isolation (OIE, 2017, Babiuk et al., 2007, House et al., 1990, Binepal et al., 2001, Salnikov et al., 2018). MDBK cells have been used to propagate the LSDV at high titres (Fay et al., 2020, Salnikov et al., 2018). Some studies also suggest use of MDBK cells for LSDV isolation (Salnikov et al., 2018). However, it is uncertain whether these are more sensitive than primary cells for LSDV isolation. We had primary goat kidney cells, primary lamb kidney cells and primary lamb testicle cells already in hand, so it’s not worth employing MDBK cells for virus isolation. However, later (as per the suggestions of the Reviewer), we also infected the MDBK cells with the clinical specimen(s) but CPE could not be observed at the first passage as it was observed in PGK (primary goat kidney) cells. This suggests that MDBK cells may not be as sensitive as PGK cells for virus isolation. However, we believe that this cannot be used as proof of principle as virus isolation from clinical specimens may be influenced by several factors such as (i) virus titre in the clinical specimens (ii) nature of cryptic (unknown) agents present in the cell culture system (iii) co infecting (unknown) agents present in the clinical specimens.

• l. 257: brackets should be deleted

Corrected as suggested (Line 256 in the revised manuscript).

• l. 295: maybe better “Serum samples from all the clinically affected animals that showed presence of…” instead of brackets

Corrected as suggested (Line 235-236 in the revised manuscript).

• l. 298: titer of 1:64 to 1:1024

Corrected as suggested (Line 238 in the revised manuscript).

• l. 308: “from India in 2019”

Corrected as suggested (Line 314 in the revised manuscript).

• l. 311: “(until this paper is being written)” is not necessary

Corrected as suggested

• l. 352: TCID50

Corrected as suggested (Line 362 in the revised manuscript).

• l. 379: phase be consistent with spelling “titer” (AE) or titre (BE) during the manuscript

“Titre” has been followed in the entire manuscript

Figures and Tables

• Figure 2: brackets not necessary

Figure 2 has been deleted in the revised manuscript

• Figure 3: could be given as supplemental information

Figure 3 has been shifted to supplementary information (S1 Fig in the revised manuscript)

• Figure 5: cold be given as supplemental information

Figure 5 has been shifted to Supplemental information (S2 Fig in the revised manuscript)

• Table 2

o footnotes (*, #) should be described

Foot notes have been described (Line 232-233 in the revised manuscript).

o information about how many scabs were taken from an individual and how many scabs were taken at the same farm would be helpful

At least 2 scabs were taken from each individual animal. The number of animals from an individual farm varies, depending on the availability of the affected animals. Please see Table 2 (Line 232-233 in the revised manuscript).

o please specify scab 22 “1 month later”, is this related to an individual animal/farm (which one?) or in general?

This is in relation to the individual animal (serial number 15). It has been mentioned in the Table 2 accordingly (Line 232-233 in the revised manuscript).

o In place of Viremia, better: LSDV genome (blood)

Viremia replaced with LSDV genome (blood) (Line 232-233 in the revised manuscript).

o Antibody titer

Corrected as suggested (Line 232-233 in the revised manuscript).

o Antibody titer should be indicated as 1:xx or transferred into ND50/ml

Corrected as suggested (written as 1:XX) (Line 232-233 in the revised manuscript).

• Figure legend Figure 2: explain red triangles

Figure 2 has been deleted in the revised manuscript

Results

• Structure outbreak and later cell culture work together – in detail:

Structured as suggested

o “reactivity of LSDV to the sera from LSDV-infected animals” should be between “Identification of the agent” and “phylogenetic analysis” as it belongs to the examination of the outbreak and results are additionally presented in Table 2

Rearranged the text as per the suggestions

Reviewer #2: 

What are the main claims of the paper and how significant are they for the discipline?

• The paper describes outbreaks of LSDV in India, and contains interesting details such as the existence of serologically positive animals which display no clinical signs, morbidity data, and phylogenetic analyses of the virus isolated. This information will be important to researchers in the field and policy makers who are trying to control the current LSD epidemic in Asia.

We thank the Reviewer for his/her enthusiasm towards this study.

Are the claims properly placed in the context of the previous literature? Have the authors treated the literature fairly?

• The literature is cited well, and most of the claims are properly placed in the context. However the utility and widespread use of MDBK cells to isolate and analyse capripoxviruses is not highlighted sufficiently.

Use of MDBK cells for LSDV isolation and propagation has been described in detail in the revised manuscript. (Line 349-362 and 368-371 in the revised manuscript).

Do the data and analyses fully support the claims? If not, what other evidence is required?

• The majority of the claims are well supported. The only unsupported claim is the use of Vero cells for the propagation of LSDV (line 367-370 and elsewhere). The authors clearly show that Vero cells cannot support propagation of field strains of LSDV (line 264-269). The virus had to be passaged 15 times in order for CPE and viral plaques to be detected on Vero cells. The literature, however, shows that MDBK cells do not require such virus adaptation and CPE can be detected on first passage.

 The authors should therefore have concluded that Vero cells are not permissive for LSDV isolation and are inferior to MDBK cells in this respect. Their conclusion that Vero cells are suitable for the propagation of LSDV is incorrect and not supported by their results.

We thank the Reviewer for this important suggestion. In fact we realised that some of our conclusions about the Vero cells were somewhat improperly written. In the revised manuscript, comparison on Vero and MDBK cell culture has been elaborated and some of the over interpretations made about Vero cells have been removed/rewritten. Also the suitability of MDBK cells for LSDV isolation and propagation has been discussed in detail. (Line 382-385 and 297-206 in the revised manuscript).

• Figure 5C is very indistinct, can the authors provide a better quality one? The plaques are not distinguishable on the current figure

Our main aim of propagating the LSDV in Vero cells was to produce plaques. Unlike the primary cells, Vero-adapted LSDV produced plaques but were of small size to capture a good image. Therefore, in the revised manuscript, we have removed Fig 5c. 

• Please provide statistical analyses for the comparisons between Vero cells and MDBK cells in Figure 6. Are the differences reported statistically significant?

We have provided statistical analysis wherever applicable. The differences in the viral titres between Vero and MDBK cells are apparently significant. However, the statistical calculation is only possible at certain time points (96hpi, 144 hpi and 196hpi) wherever there are variations in the biological replicates. The absence of any variations in three the biological replicates limits the calculation of significant statistical difference. 

However, we have provided the statistical analysis for the levels of viral DNA in Vero and MDBK cells at different time points. Pair-wise statistical comparisons were performed using Student’s t test. (Fig 3 and Line 542-548 in the revised manuscript).

• Please re-examine the legend for Figure 6. It is not correct. A and B are growth curves. C and D (not currently mentioned in the legend) are PCRs.

We thank the Reviewer for pointing out this mistake. The legend has been corrected (Line 528-534 in the revised manuscript).

• Make it clear in the legend and text for Figure 6 that the data presented is for the cell culture adapted strain of LSDV, not a field strain. This is very important for diagnosticians who may read the manuscript and not realise straight away that Vero cells do not support LSDV isolation.

We have mentioned in the Figure legend and discussion section that this is applicable to cell culture adapted strain of LSDV (Line 190, 378, 388 and 529 in the revised manuscript).

• Do not use PFU/ml as a synonym for TCID50, as described in the legend of Figure 6. They are very different things. If you counted plaques (or foci) then use PFU. If you examined cells for CPE, use TCID50.

We thank the Reviewer for pointing out this issue. Virus titres were determined by TCID50 assay and converted to estimated plaque-forming units (PFU) by the conversion TCID50 ≈ 0.7 PFU as per the standard conversion formula described previously by Coves-Datson EM et al (2020, PNAS, USA). This was carried out to ensure homogeneity in the expression of viral titres (PFU/ml or TCID50/ml).

PLOS ONE encourages authors to publish detailed protocols and algorithms as supporting information online. Do any particular methods used in the manuscript warrant such treatment? If a protocol is already provided, for example for a randomized controlled trial, are there any important deviations from it? If so, have the authors explained adequately why the deviations occurred?

• Not applicable

We thank the Reviewer

If the paper is considered unsuitable for publication in its present form, does the study itself show sufficient potential that the authors should be encouraged to resubmit a revised version?

• Not applicable

We thank the Reviewer

Are original data deposited in appropriate repositories and accession/version numbers provided for genes, proteins, mutants, diseases, etc.?

• The LSDV strain has been deposited in a repository (lines 261-262) although this could not be confirmed on the website of the depository

We have asked the concerned repository to update the information of our deposited LSDV strain. In fact the information has been updated. Please visit at www.ncvtc.org.in (Under Distribution of Microbes)

Are details of the methodology sufficient to allow the experiments to be reproduced?

• Yes

We thank the Reviewer

Is the manuscript well organized and written clearly enough to be accessible to non-specialists?

• Yes, it is very nicely written.

We thank the Reviewer

---

## [Decision Letter · Decision Letter 1]

22 Dec 2020

PONE-D-20-31124R1

Isolation and characterization of lumpy skin disease virus from cattle in India

PLOS ONE

Dear Dr. Kumar,

Thank you for submitting your manuscript to PLOS ONE. After careful consideration, we feel that it has merit but does not fully meet PLOS ONE’s publication criteria as it currently stands. Therefore, we invite you to submit a revised version of the manuscript that addresses the points raised during the review process.

As stated by the reviewer 2, the usage of VERO cells to isolation and propagation of wild type LSDV is still a mater of debate. Thus you have to clearly indicate as requested that adaptation to this cell line is needed to allow usage of VERO cells until you were able to shown multiple isolation of LSDV from field sampling using this cell line and not the classical MDBK cells. The fact that amplification of a reference adaptated LSDV strain in VERO can be done is of interest but the limitation of such a strain remains to be highlighted to people that are more interested in the epidemiological aspect of the LSDV emergence in India to avoid any missinterpretation.

We look forward to receiving your revised manuscript.

Kind regards,

Pierre Roques, Ph.D.

Academic Editor

PLOS ONE

Reviewers' comments:

Reviewer's Responses to Questions

**Comments to the Author**

1. If the authors have adequately addressed your comments raised in a previous round of review and you feel that this manuscript is now acceptable for publication, you may indicate that here to bypass the “Comments to the Author” section, enter your conflict of interest statement in the “Confidential to Editor” section, and submit your "Accept" recommendation.

Reviewer #1: All comments have been addressed

Reviewer #2: (No Response)

2. Is the manuscript technically sound, and do the data support the conclusions?

Reviewer #1: Yes

Reviewer #2: Partly

3. Has the statistical analysis been performed appropriately and rigorously? 

Reviewer #1: Yes

Reviewer #2: Yes

4. Have the authors made all data underlying the findings in their manuscript fully available?

Reviewer #1: Yes

Reviewer #2: Yes

5. Is the manuscript presented in an intelligible fashion and written in standard English?

Reviewer #1: Yes

Reviewer #2: Yes

6. Review Comments to the Author

Reviewer #1: The performed changes in the revised manuscript are fine for me. This revised version is acceptable for publication.

Reviewer #2: The authors have addressed the majority of the comments by the reviewers. However they need to make very clear throughout the manuscript that they are analysing their laboratory adapted strain of LSDV in figure 3, and not a wildtype strain. The inability of Vero cells to support propagation of LSDV needs to be clearly stated. The authors also need to make clear that MDBK cells are a suitable and widely used cell line for LSDV isolation and propagation. This information is very important for diagnosticians who may read the manuscript and not realise straight away that Vero cells do not support LSDV isolation. Thus please take in account the comments from the reviewer 2.

Please correct “LSDV” to “Vero-cell adapted strain of LSDV” on line 35, 39, 292, 295, 298, 302, 394, 398 (twice) and 536. Also in the legend to Figure 3.

Add MDBK cells to the list of cells used for LSDV isolation on line 349-350.

Delete lines 379-380 and 396 which state that Vero cells are suitable for the propagation of LSDV. They are clearly not.

Line 95-96 Delete the statement “However, a plaque assay to precisely quantify infectious LSDV is still lacking.” This plaque assay has been described in Fay et al 2020.

Figure S2 (b) the images are out of focus, which precludes examination of the CPE at p15. Do the authors have a better image?

Figure S2 (b) LSDV is spelt incorrectly in the figure heading.

7. PLOS authors have the option to publish the peer review history of their article (what does this mean?). If published, this will include your full peer review and any attached files.

Reviewer #1: No

Reviewer #2: No

---

## [Author Response · Author response to Decision Letter 1]

23 Dec 2020

To,

Pierre Roques, Ph.D.

Academic Editor

PLOS ONEPLOS One

December 23, 2020

Subject: Submission of revised manuscript for publication

Dear Dr. Roques,

We are indebted for your help to facilitate the review of our manuscript entitled “Isolation and characterization of lumpy skin disease virus from cattle in India”. We wish to thank the reviewers for their careful evaluation of our manuscript.

Nowhere in the entire manuscript had we described that Vero cells may be used for virus isolation/propagation of wild-type LSDV strain. We just described that LSDV, in fact can be adapted to Vero cells and adapted virus can be grown in high titers in Vero cells. 

Phillipa M. Beard’s group (Fay et al., 2020) has published a paper about propagation of LSDV in MDBK cells which has been cited by us as well. Although MDBK cells have been used for LSDV propagation but there is no mention in the literature that MDBK cells are better than primary cells for LSDV isolation. Therefore, we used primary cells (primary goat kidney cells) for LSDV isolation in this study. 

After isolation in primary cells, we adapted the virus in Vero cells with two major objectives (i) Attempting whether Vero adapted LSDV can form plaques because a plaque assay for LSDV has not yet been described (ii) The current LSDV vaccine is based upon attenuation of the LSDV in primary cells/MDBK cells but it has disadvantage of producing a local reaction at the site of inoculation. Therefore, we hypothesized that attenuation in a different cell line (such as Vero cell used by us) may overcome with this problem. However, testing vaccine efficacy of Vero cell-adapted LSDV strain needs further investigations.

In the previous version of the manuscript, we had already described (Line number 357-361) that MDBK cells are also sensitive for LSDV isolation. Similarly, we had clearly mentioned that Vero cells are not suitable for virus isolation from the clinical specimens (Line number 382-385). Raising this issue again and again suggests that Reviewer 2 has either not read the revised manuscript carefully or has conflict of interest with this study. 

All the corrections in the revised manuscript are highlighted in red. Our responses to the reviewer’s specific comments are outlined below in bold letters. We hope that you and the reviewers will find this revised version acceptable for publication in your esteemed Journal. 

Yours sincerely

Naveen Kumar, Ph.D. 

Principal Scientist (Veterinary Virology)

National Center for Veterinary Type Cultures

Sirsa Road, Hisar, Haryana-125001

India

email: naveenkumar.icar@gmail.com

Reviewer #1: 

The performed changes in the revised manuscript are fine for me. This revised version is acceptable for publication.

We thank the Reviewer for considering our manuscript acceptable for publication

Reviewer #2: 

The authors have addressed the majority of the comments by the reviewers. However they need to make very clear throughout the manuscript that they are analysing their laboratory adapted strain of LSDV in figure 3, and not a wild-type strain. 

We have mentioned “Vero cell-adapted LSDV” instead of LSDV wherever it’s required (please see line number 35, 40, 190, 292, 295, 296, 388, 396, 398, 528-529 in the revised manuscript). 

The inability of Vero cells to support propagation of LSDV needs to be clearly stated. 

Even in the previous version of the manuscript we had clearly mentioned (Line number 382-385 in the current and previous version of the manuscript) that Vero cells are not suitable for LSDV isolation from the clinical specimens.

The authors also need to make clear that MDBK cells are a suitable and widely used cell line for LSDV isolation and propagation. This information is very important for diagnosticians who may read the manuscript and not realise straight away that Vero cells do not support LSDV isolation. Thus please take in account the comments from the reviewer 2.

In the previous version of the manuscript, we had already described (Line number 357-361 in the current and previous version of the manuscript) that MDBK cells are also sensitive for LSDV isolation. We never compared Vero cells with MDBK/primary cells for virus isolation. We just compared the growth of Vero cell-adapted LSDV in Vero and MDBK cells (Fig. 3) where Vero cells were certainly found better than MDBK cells for the growth of LSDV (Vero cell-adapted). 

Please correct “LSDV” to “Vero-cell adapted strain of LSDV” on line 35, 39, 292, 295, 298, 302, 394, 398 (twice) and 536. Also in the legend to Figure 3.

We have mentioned “Vero cell-adapted LSDV” instead of LSDV wherever it’s required (please see line number 35, 40, 190, 292, 295, 296, 388, 396, 398, 528-529 in the revised manuscript). 

Add MDBK cells to the list of cells used for LSDV isolation on line 349-350.

Again, in the previous version of the manuscript, we had already described (Line number 357-361 in the current and previous version of the manuscript) that MDBK cells may also be used for LSDV isolation.

Delete lines 379-380 and 396 which state that Vero cells are suitable for the propagation of LSDV. They are clearly not.

We totally disagree with the Reviewer about deleting this statement because this is about the growth of Vero cell-adapted LSDV (not wild type LSDV) where Vero cells produced higher titer than MDBK cells. 

Line 95-96 Delete the statement “However, a plaque assay to precisely quantify infectious LSDV is still lacking.” This plaque assay has been described in Fay et al 2020.

The paper by Beard’s group (Fay et al., 2020) describes focus forming assay. Unlike plaque assay, the foci (formed in this assay) need to be counted under microscope, thereby it is not considered as good as plaque assay where plaques can be directly counted with naked eyes. 

Figure S2 (b) the images are out of focus, which precludes examination of the CPE at p15. Do the authors have a better image?

We have provided a different better image

Figure S2 (b) LSDV is spelt incorrectly in the figure heading.

Corrected

---

## [Editor Report · Decision Letter 2]

23 Dec 2020

Isolation and characterization of lumpy skin disease virus from cattle in India

PONE-D-20-31124R2

Dear Dr. Kumar,

We’re pleased to inform you that your manuscript has been judged scientifically suitable for publication and will be formally accepted for publication once it meets all outstanding technical requirements.

Kind regards,

Pierre Roques, Ph.D.

Academic Editor

PLOS ONE

Additional Editor Comments (optional):

thank you for your comments and have a good new year and a happy Christmas
---

## [Editor Report · Acceptance letter]

28 Dec 2020

PONE-D-20-31124R2 

Isolation and characterization of lumpy skin disease virus from cattle in India 

Dear Dr. Kumar:

I'm pleased to inform you that your manuscript has been deemed suitable for publication in PLOS ONE. Congratulations! Your manuscript is now with our production department. 

Kind regards, 

on behalf of

Dr. Pierre Roques 

Academic Editor

PLOS ONE